# Impact of Combined Thermo- and Photo-Oxidation on the Physicochemical Properties of Oxo-Biodegradable Low-Density Polyethylene Films

**DOI:** 10.3390/polym17020193

**Published:** 2025-01-14

**Authors:** M. F. Rojas-Trejo, A. Valadez-Gonzalez, L. Veleva, R. Benavides, M. T. Rodriguez-Hernandez, M. V. Moreno-Chulim

**Affiliations:** 1Unidad de Materiales, Centro de Investigación Científica de Yucatán, Calle 43 No. 130 Col. Chuburná de Hidalgo, Merida 97205, Mexico; mafer.rojas.trejo@gmail.com (M.F.R.-T.); veromor@cicy.mx (M.V.M.-C.); 2Applied Physics Department, Center for Research and Advanced Studies (Cinvestav-Mérida), Merida 97310, Mexico; 3Centro de Investigación en Química Aplicada (CIQA), Blvd. Enrique Reyna Hermosillo 140, Saltillo 25294, Mexico; roberto.benavides@ciqa.edu.mx (R.B.); teresa.rodriguez@ciqa.mx (M.T.R.-H.)

**Keywords:** oxo-degradable LDPE, thermo- and photo-oxidative treatments, abiotic degradation

## Abstract

This research addresses the study of the combined effect of two abiotic treatments, a thermo-oxidative treatment followed by a photo-oxidative treatment with ultraviolet light, on the physicochemical properties of commercially available low-density polyethylene films with an oxo-degradant additive (OXOLDPE) and without (LDPE). The change in the oxidized film properties was characterized using FTIR, XRD, TGA, GPC, and SEM analytical techniques. The results indicated that the increment in carbonyl index (CI) and crystallinity percentage (X_XRD_) was higher for those films that received the combined oxidative treatments compared with those that received only one of them, thermo- or photo-oxidative treatment. Moreover, the combined oxidative treatments produced more ester and carboxylic groups on the degradation products than the other single treatments. An analysis of variance (ANOVA) was carried out, and a synergistic effect was observed between the thermo- and photo-oxidative treatments for both ester and carboxylic degradation products. TGA results revealed that the loss of thermal stability in the films was more significant after their exposure to the combined thermo- and photo-oxidative treatments compared with those which received only one. The GPC results showed that the combined oxidative treatment is necessary to decrease the Mz and M_z+1_ average molecular weight of degraded films containing an oxo-degradant additive to the same extent as MW and Mn. The SEM surface appearance of the films changed more drastically after their exposure to the combined thermo- and photo-oxidative treatments, and they seemed to erode with the presence of inorganic fillers (CaCO_3_). These results suggest that the combined oxidative treatments produced degradation products with lower molecular weight and greater content of ester and carboxylic groups that should enhance its environmental biodegradability.

## 1. Introduction

The accumulation of plastics in the environment continues to increase at ever higher rates, mainly because they are materials with very low biodegradation rates. In contrast, the manufacture of plastic materials increases year after year. Polyolefins represent a significant fraction of the synthetic plastics manufactured annually, and low-density polyethylene (LDPE), in particular, stands out among them because most of products composed of it are single-use [1,2]. To increment the biodegradability of plastics once their useful life has ended, additives based on stearates of transition metals called oxo-degradants are added during their manufacture. Incorporating oxo-degradant additives can accelerate the degradation process of LDPE since transition metals can act as catalysts for photo- and thermal-degradation processes. The catalytic degradation of polyethylene in the presence of transition metals has been attributed to its ability to generate free radicals on its surface. These radicals later react with oxygen to generate low-molecular-weight oxidation products as oxygen is introduced into the carbon chain as hydroxyls, carbonyls, and peroxides [3,4,5]. The stearates of Fe^3+^, Mn^2+^, or Co^2+^ are the most commonly used oxo-degradants, while Ca^2+^ and Cu^2+^ stearates are used with less frequency [3,4,5,6,7,8]. These materials are called oxo-biodegradable, but there is currently controversy over whether oxo-biodegradable plastics are actually biodegradable and usable as a carbon source by microorganisms. Many studies confirm that oxo-biodegradable plastics are biodegradable [9,10,11,12,13,14], while many others consider that they are not [15,16,17,18,19,20]. This discrepancy has even led the EU to prohibit the use of such materials [21]. In addition, several studies indicate that oxo-biodegradable plastics tend to generate a more significant amount of microplastics when they degrade in the environment compared to conventional plastics because the formation of these microparticles is closely related to the cleavage of the polymeric chains, the chemical structure of the degradation products formed and the increase in the percentage of crystallinity [22,23,24]. It is known that biodegradation of a synthetic polymer is a process that consists of two stages, biotic and abiotic degradation, that lead to the formation of low molecular weight fragments bearing ester and carboxylic groups that can be used as a carbon source by microorganisms [9,11,12,25,26,27,28,29,30]. Likewise, it is known that the products of thermo-oxidative degradation or photo-oxidative degradation of polyethylene give rise to shorter polymer chains (decrease in molecular weight) bearing ester and carboxylic groups. The vast majority of research on the abiotic degradation of oxo-biodegradable polyethylene has separately studied the influence of thermo-oxidative degradation [31,32,33,34,35,36,37,38,39] or photo-oxidative degradation [40,41,42,43,44,45,46] on its physicochemical properties and/or its biodegradability. The results of these studies have shown that although the decrease in molecular weight and the presence of ester and carboxylic groups does indeed increase the rate and quantity of LDPE assimilated by microorganisms, these are still low. Moreover, what has been regularly observed is that microorganisms can only metabolize short oxidized molecular chains. These findings highlight how crucial the abiotic degradation stage is in the biodegradation of plastics, such as LDPE, with and without oxo-degradant additives.

This work aims to study the combined effect of thermo-oxidative and photo-oxidative degradation on the physicochemical properties of commercial LDPE films with and without an oxo-degradant additive.

For this purpose, commercial low-density polyethylene films with an oxo-degradant additive (OXOLDPE) and without one (LDPE) were subjected to different oxidative treatments: 90 days at 70 °C, 1000 h of UV irradiation, and subsequently a combination of both treatments. The FTIR, XRD, TGA, GPC, and SEM-EDS analytical techniques were used to assess the changes in the films’ physical and chemical properties.

## 2. Materials and Methods

### 2.1. Materials

Commercial low-density polyethylene film (LDPE, 0.035 mm thickness) and an oxo-biodegradable LDPE film (OXOLDPE, 0.035 mm thickness) were obtained from a local producer (POLIMERIDA, YUC, MX). Specimens (100 mm × 150 mm × 0.035 mm) were cut, conditioned at 25 °C and 25% RH, and kept in sealed bags until exposure to experimental tests.

### 2.2. Experimental Tests

#### 2.2.1. Thermo-Oxidative Test

The samples in triplicate were exposed to a thermo-oxidative environment in a forced air convection oven (EQUATHERM, Seattle, WA, USA) at 70 °C for 90 days. Then, they were withdrawn from the oven and conditioned at 25 °C and 25% RH and kept in sealed bags until exposure to experimental tests.

#### 2.2.2. Photo-Oxidative Test

The samples in triplicate were exposed to an accelerated UV aging camera (QUV-SPRAY, Westlake, OH, USA) and continuously irradiated with UVB-313 lamps (0.71 W/m^2^) at 60 °C for 1000 h. Subsequently, they were conditioned at 25 °C and 25% RH, and stored in sealed bags in a desiccator for later analysis.

#### 2.2.3. Combined Thermo- and Photo-Oxidative Test

In order to assess the combined effect of both oxidative tests, the tested samples were previously thermo-oxidized (at 70 °C for 90 days) and posteriorly irradiated (1000 h, at 0.71 W/m^2^ and 60 °C) with a UV aging camera. After that, the specimens were conditioned at 25 °C and 25% RH in a desiccator and stored in sealed bags for later analysis. The tested samples were coded as indicated in Table 1.

### 2.3. Physicochemical Characterization

#### 2.3.1. Fourier Transformed Infrared Spectroscopy (FTIR)

The FTIR characterization was performed using a Nicolet 8700 Thermo Scientific spectrometer (Thermo Electron Scientific^®^ Instruments LLCD, Madison, WI, USA). The spectra were recorded with a resolution of 2 cm^−1^ and 100 scans in a wavenumber range from 4000 cm^−1^ to 500 cm^−1^. The spectra obtained were normalized using the band at 2020 cm^−1^ as internal standard before they were analyzed [47,48]. The overall carbonyl index (CI) was calculated (Equation (1)), according to the method proposed by Almond et al. [49]. The interval of the FTIR spectra of carbonyl peaks from 1850 to 1650 cm^−1^ was integrated and normalized with the methylene (CH_2_) absorption peak integrated area in the range of 1500–1420 cm^−1^.(1)$$Carbonyl\;Index\;\left(IC\right)=\frac{Area\;of\;the\;arbsorption\;peaks\;of\;carbonyls(1850-1650\;{\mathrm{cm}}^{-1})}{Area\;of\;the\;methylene\;absorption\;peak\;\;(1500-1420\;{\mathrm{cm}}^{-1})}$$

The carbonyl region was deconvoluted using the Lorentzian non-linear least squares fitting procedure (Origin Pro 9, OriginLab Corporation, Northampton, MA, USA) to calculate the esters (COOR) and carboxylic acids (COOH) contribution. The ester and carboxylic acid indexes were calculated using the peaks at 1740 cm^−1^ and 1710 cm^−1^, normalized with the methylene absorption peak, respectively, like CI. The unsaturation (C=C) index was calculated using the peak at 990 cm^−1^ and also normalized with the methylene absorption peak, like CI.

#### 2.3.2. X-Ray Diffraction (XRD)

A Bruker D-8 Advance diffractometer (Billerica, MA, USA) operating at 40 kV and an intensity of 30 mA (Kα of Cu at λ = 1.5406 Å, 2θ = 15–25° (0.02° step size)) and a step time of 0.5 s was used to obtain the XRD spectra of the studied samples. The percentage of crystallinity was calculated using the following relation (2):(2)$$X_{XRD}=\frac{A_{c}^{110}+A_{c}^{200}}{A_{a}+A_{c}^{110}+A_{c}^{200}}\times 100$$
where A*a*, A*c*^100^, and A*c*^200^ are the areas under the amorphous halo, the 100 and the 200 reflections, respectively. The mean crystallite size (L_110_), corresponding to 110 cell planes, was obtained by XRD data and the Scherrer Equation (3) [44].(3)$$L_{110}=\frac{K\lambda }{\beta \cos\theta }$$
where *L*_110_ represents the mean dimension of the crystal normal to the corresponding 110 plane, *K* is a first-order constant, *λ* represents the wavelength, 2θ is the scattering angle, and β is the half-height of the scattering peak. In this study, the parameters λ = 1.5418 Å, 2θ = 21.5° were used.

#### 2.3.3. Thermogravimetric Analysis (TGA)

The thermal stability of the studied sample films was assessed using a TGA-7 Perkin-Elmer Inc. (Waltham, MA, USA). The sample (10 mg) was heated at a constant rate of 5 °C/min from 30 °C to 600 °C, under a nitrogen atmosphere.

#### 2.3.4. Gel Permeation Chromatography (GPC)

The molecular weight distributions (MWD) of LDPE and OXOLDPE films were evaluated before and after the thermo- and photo-oxidative treatments using an Agilent PL-GPC 220 High Temperature Chromatograph (Santa Clara, CA, USA) at 140 °C. For this, 20 mg of the sample was weighed and dissolved in 15 mL of 1,2,4-Trichlorobenzene for 1 h at 170 °C. The solution was then filtered through stainless steel filters (0.5 microns).

#### 2.3.5. Scanning Electron Microscopy (SEM-EDS)

The surface morphology changes and elemental analysis of samples before and after being tested were evaluated with SEM-EDS (JEOL Model JSM- 7600F (Akishima, Japan), at 15 kV). The micrographs were obtained at a magnification of x500. The specimens were previously coated with a thin layer of gold/palladium.

## 3. Results and Discussions

### 3.1. Fourier Transformed Infrared Spectroscopy (FTIR)

The effect of thermo-oxidative and photo-oxidative treatments, and the combination of both, on the chemical properties of the studied films can be observed in the FTIR spectra (Figure 1). According to the absorbance intensity of the peaks, the degradation of the LDPE and OXOLDPE films due to the thermo-oxidative treatment is lower than that induced by the photo-oxidative treatment and much lower than the combined treatment. This is evident from the carbonyl group region (C=O) peaks between 1600–1800 cm^−1^ and in the hydroxyl group region (–OH) between 3200–3600 cm^−1^. The formation of these groups has been reported as a consequence of the oxidative degradation of polyolefins [50,51,52,53,54]. Likewise, the absorbance bands of the OXOLDPE films seem to be higher than those of the LDPE. These facts suggest that films containing the oxo-degradant additive have experienced more significant oxidative degradation, which is consistent with what has been reported in the literature [3,8,15,18,31,38,39,40].

The CI values, calculated using Equation (1) for LDPE and OXOLDPE films due to the oxidative treatments, are presented in Figure 2. As can be observed, the presence of degradation products containing carbonyl groups increases in both types of films in the following order: (without treatment) < thermo < photo << thermo+photo. However, it should be noted that the content of these carbonyl groups is higher in the films containing the oxo-degrading additive (OXOLDPE). It is also evident that the increase in carbonyl group content due to the combined oxidative treatment is higher than expected from an additive behavior. These facts suggest that there is a synergistic effect between the thermo- and photo-oxidative treatments for the formation of degradation products containing carbonyl groups for both LDPE and OXOLDPE.

This behavior is interesting since it tells us that the combination of thermo- and photo-oxidative treatments seems to be more relevant than the presence of the oxo-degradant additive.

The Analysis of Variance (ANOVA) was conducted using the MINITAB^TM^ 18 statistical software (Minitab, LLC., State College, PA, USA) to clarify this. In Figure 3a, the Pareto Chart shows that the single oxidative treatments, their combination, and the kind of film (LDPE or OXOLDPE) are all statistically significant (*p* < 0.05) in the formation of carbonyl groups, though the photo-oxidative treatment contributes the most to it. It should be noted that the contribution of the combined treatments is higher than the presence of the oxo-degradant additive.

In Figure 3b, it can be observed that the carbonyl group formation is more sensitive to both single treatments than the kind of films, i.e., the treatment type contributes to a greater extent than the presence of the oxo-degradant additive. Figure 3c shows the interaction graphs of the thermo- and photo-oxidative treatments. It can be observed that the thermo-oxidative treatment drastically increases the effect of the photo-oxidative treatment on the formation of carbonyl groups. This behavior suggests that there is a synergistic effect between both oxidative treatments [55].

The carbonyl region was deconvoluted using the Lorentzian non-linear least squares fitting procedure (Origin Pro 9) to calculate the esters (COOR) and carboxylic acids (COOH) contribution. Their corresponding specific carbonyl group indexes was calculated as described in Section 2.3.1.

Figure 4 shows the corresponding carbonyl indexes for ester and carboxylic groups that bear the film’s degradation products. It can be seen in Figure 4a that the ester group formation predominates in the degradation products for thermo-, photo- and the combined oxidative treatments for LDPE. On the other hand, for OXOLDPE, the formation of carboxylic groups is enhanced by the combined oxidative treatments, as can be seen in Figure 4b. The formation of ester groups and carboxylic groups during thermo-oxidative and photo-oxidative degradation in polyethylene with and without the presence of oxo-degrading additives has been reported by several researchers [7,11,33,39,52,56,57,58,59,60]. The results found in this work suggest that the combination of oxidative treatments in the presence of an oxo-degrading additive promotes the generation of carboxylic groups over that of ester groups.

The results of the ANOVA analysis, using the Minitab 18 software, for the carboxylic and ester carbonyl group indexes are shown in Figure 5 and Figure 6, respectively.

In Figure 5, it can be seen that thermo-, photo-, the combined oxidative treatments and the oxo-degradant additive (OXOLDPE) are statistically significant. In Figure 5b, it can be seen that the photo-oxidative treatment is more important than the thermo- one, and than the oxo-degradant additive. The synergy between both oxidative single treatments is evident in Figure 5c.

In Figure 6, it can be seen that the single oxidative treatments and their combination, but not the oxo-degradant additive, are statistically significant in the ester group formation. In Figure 6a, it can be seen that the photo-oxidative treatment contributes to a great extent compared to thermo-oxidative treatment in ester formation. In Figure 6c, it is evident that there is a synergistic effect between both single oxidative treatments.

### 3.2. X-Ray Diffraction (XRD)

Table 2 shows the crystallinity (X_XRD_) percentage and crystal size (L_110_) for LDPE and OXOLDPE films corresponding to the oxidative treatments. The results indicate that the thermo-oxidative treatment increased the crystallinity fraction by about 20% for both LDPE and OXOLDPE films compared with the films without treatment. On the other hand, the photo-oxidative treatment increased the percentage of LDPE crystallinity by 20%, and 60% for the OXOLDPE films. Likewise, the combined oxidative treatment increased the crystallinity fraction of the LDPE and OXOLDPE films by 40% and 75%, respectively. These facts suggest that the formation of ester and carboxylic acid groups, shown with the FTIR results, led to the scission of the polymeric chains [3,35,39,46]. Gedde et al. [61] considered that chain scission occurs essentially within the amorphous phase, favoring the release of short-chain segments. These segments, having greater mobility than the initial chains, would have the ability to migrate to the surface of the crystalline phase and participate in secondary crystallization (chemical crystallization).

As for the average size of the crystals (L110), there is not much difference between the results obtained for LDPE and OXOLDPE films. It seems that incorporating the oxo-degradant additive in LDPE does not affect the average size of the formed crystallites. As regards the effect of the oxidative treatments, it is evident that the sizes of the crystallites are very similar regardless of the type of oxidative treatment to which the films were subjected (Table 2). These facts suggested that a homogeneous crystallization process prevailed over time, in addition to forming mobile small chain fragments from the chain scission reaction, subjected to reorganization and re-crystallization, while the existing crystallites did not break apart with increasing time [62].

### 3.3. Thermogravimetric Analysis (TGA)

The thermograms for LDPE and OXOLDPE before and after the oxidative treatments are shown in Figure 7, where it can be observed how the thermal stability of the films decreases with the different oxidative treatments.

Figure 7a reveals that the mass loss of both LDPE and OXOLDPE films started at low temperatures, and after different oxidative treatments their thermal stability decreased because of the abiotic degradation process. For both films, the loss of thermal stability followed the order: thermo+photo>>photo>thermo. This behavior agrees with the FTIR results (Figure 1), which reveal a more significant formation of oxidized degradation products after the combined oxidative treatment for both films. Meanwhile, the mass loss rate as a function of temperature (Figure 7b) shows that the mass loss started around 160 °C for the films subjected to photo-oxidative treatment and those subjected to combined treatment (thermo+photo). This behavior suggests that the products of oxidative degradation have a much lower molecular weight than those formed after the thermo-oxidative treatment of the studied films [19,62,63,64,65]. Table 3 compares the temperatures at which 10% and 25% mass losses occurred (T_10%_, T_25%_) and the temperature (T_max_) at which the mass loss rate (dm/dT) was at its maximum.

The results show that the mass loss temperatures of 10% and 25% decrease for both LDPE and OXOLDPE films with the oxidative treatments in the following order: thermo+photo>photo>thermo. This is an indication that there is a decrease in the molecular weight of the films due to the oxidative treatments [4,64,65]. On the other hand, the behavior of T_max_ is different. For LDPE films, T_max_ remained constant after the thermo- and photo-treatments, and increases by about 14 °C after the combined treatment, probably due to the increase in the percentage of crystallinity of the films [62,66,67]. For OXOLDPE films, T_max_ increased by about twice (27 °C) after the photo-treatment with respect to the LDPE films. However, for the films that received the combined oxidative treatment, the increase in T_max_ was 17 °C, even though the percentage of crystallinity of these samples was much higher than those that received only the photo-treatment. These facts suggested that the degradation products formed are less thermally stable than those formed after the combined oxidative degradation process, compared to the photo-oxidative one.

### 3.4. Gel Permeation Chromatography (GPC)

Figure 8 shows the molecular weight distributions obtained by GPC of LDPE and OXOLDPE after being subjected to the different oxidative treatments.

This figure shows that the thermo-oxidative treatment did not modify significantly the molecular weight distribution (MWD) of any films. On the other hand, the shift towards the region of lower molecular weight is evident after the photo-oxidative and combined oxidative treatments of the films. Several researchers have published similar results, mainly after photo-oxidative treatments of LDPE films with and without oxo-degradant additives [4,12,19,31,32,33,53,54]. The inserted box in Figure 8 shows that the photo-oxidative treated films present a shoulder around 105 (g/mol). This shoulder practically disappears in those films that received the combined oxidative treatment. This finding is very interesting, since it indicates that the combined treatment decreases the average molecular weight of the LDPE and OXOLDPE films and drops the fraction of higher molecular weights. Table 4 summarizes the average molecular weights in number (M_n_), weight (M_W_), and viscosity (M_V_) of the films, together with their polydispersity index (M_W_/M_n_), which is frequently used to characterize the MWD of polymeric matrices, as an indicator of its amplitude. Table 4 also incorporates the average molecular weights M_z_ and M_z+1_, the values of which are more influenced by the existence of polymer chains of higher molecular weight, compared to the M_n_ and M_w_ average weights, calculated from the molecular weight distributions presented in Figure 8. From the point of view of processing and performance of the material during its useful life, the importance of the molecular averages M_z_ and M_z+1_ have been pointed out [67,68,69,70], but not from the point of view of the material degradation, once its useful life has expired. What is widely documented is that during its biodegradation, microorganisms are only able to use as substrate the fractions that contain short molecular chains, less than 5000 g/mol.

The results presented in Table 4 suggest that the thermo-oxidative treatment does not significantly modify the values of M_n_ and M_W_ and, therefore, the M_W_/M_n_ ratio of either LDPE or OXOLDPE films. Regarding the photo-oxidative treatment, the changes are more drastic: the values of M_n_, M_W_ and M_V_ fall abruptly for both films, while the polydispersity doubles for LDPE and triples for OXOLDPE, i.e., the fraction of high molecular weight chains remained significant, according to the values of M_z_ and M_z+1_. On the other hand, the effect of the combined oxidative treatment does not significantly affect the values of M_n_ and M_w_, compared to those after the photo-oxidative film treatment. However, the polydispersity decreased significantly from 8.4 to 3.9 for LDPE and from 9.6 to 5.3 for OXOLDPE. To appreciate these changes in more detail, the percentages of variation of these indicators were calculated by taking the values of the LDPE films as a reference.

Figure 9 compares the change in molecular weights of the tested LDPE and OXOLDPE films. The comparison of the molecular weights confirms that the photo- and combined oxidative treatments significantly reduced the values of M_z_ and M_z+1_ for the tested films, because of the reduction of the fraction of chains having a higher molecular weight. For LDPE films, the most significant change is in Mz+1, which increased from 75% after the photo-oxidative treatment to 95% during the combined treatment.

It seems that this effect is less pronounced in the presence of oxo-degradant additives in OXOLDPE (Figure 9b). The M_z_ and M_z+1_ of the OXOLDPE films decreased by 41% and 61%, respectively, after the photo-oxidative treatment, while with the combined treatment, the values of M_z_ and M_z+1_ decreased by 85% and 86%, respectively.

### 3.5. Scanning Electron Microscopy (SEM-EDS)

SEM images of the change in the surface morphology of the OXOLDPE films subjected to different degradation treatments are presented in Figure 10.

The reference OXOLDPE surface (Figure 10a) was relatively smooth and white spots (zones A) were visible, which EDS analysis suggested to be calcium (Ca). The surface of the thermo-oxidized specimen (at 70 °C for 90 days, Figure 10 b) revealed the white Ca-spots, as maintained on the surface. After the photo-oxidative treatment (for 1000 h, Figure 10c), cracks appeared on the film surface because of the crystallinity increase and reduction in the molecular mass [19,28,37,38,65]. The morphology of the films treated with the combined thermo- and photo-oxidative treatments (Figure 10d) changed drastically, i.e., the surface was more eroded and exhibited white spots in different shapes (zones A); EDS analysis confirmed that the films of OXOLDPE had been filled with inorganic filler grains, mainly of Ca and Mg at a very low content. The Ca filler (up to 40%) has been considered as the main oxo-degradant additive (as CaCO_3_) for the LDPE films. The calcium carbonate (CaCO_3_) filler is of low cost and may improve the films’ structural stability during processing. It has been reported to act as a weathering barrier protective agent against UV light incidence [51].

The LDPE and OXOLDPE films, subjected to thermo- and photo-oxidative treatments as well as to the combination of both treatments, were characterized by means of FTIR, XRD, TGA and GPC analysis. The results revealed that the combined oxidative treatment resulted in the formation of oxidized degradation products with a higher abundance of ester and carboxylic groups, as well as in a greater drop in the average molecular weights M_z_ and M_z+1_, compared to those obtained after individual thermo- and photo-oxidative treatments. The effect of the combined oxidative treatment was evident for the LDPE and OXOLDPE films tested, and it was suggested that there was a synergistic effect between both oxidative treatments when the combined test was carried out. It is important to highlight that because of the presence of the oxo-degrading additive, the formation of carboxylic groups was greater than that of ester groups.

These results are very interesting, and merit further study, because they provide the guidelines for achieving higher percentages of mineralization of LDPE films and even very probably of other types of polyolefins.

## 4. Conclusions

The study of the combined effect of two abiotic treatments (a thermo-oxidative treatment followed by a photo-oxidative treatment with ultraviolet light) on the physicochemical properties of commercially available low-density polyethylene (LDPE) films and with oxo-degradant additives (OXOLDPE) was carried out. The results indicate that the increment in carbonyl index (CI) and crystallinity percentage (X_XRD_) was higher for those films that received the combined oxidative treatments, compared with those that received only one treatment, either thermo- or photo-oxidative. Moreover, the combined oxidative treatment produced more ester and carboxylic groups as degradation products than single treatments of the tested films. Deconvolution in the carbonyl band indicated a high concentration of ester groups and carboxylic acids, suggesting that the Norrish I reaction mechanism predominates during the cleavage of the polymer chains. An analysis of variance (ANOVA) was carried out and showed that there was a synergistic effect between the thermo- and photo-oxidative treatments for the ester and carboxylic degradation products. TGA results showed that the loss of thermal stability in the tested films was more significant after their combined thermo- and photo-oxidative treatments, compared with that observed after single treatments. The GPC analysis showed that the combined oxidative treatment led to decrease of the M_z_ and M_z+1_ average molecular weights of the degraded films containing an oxo-degradant additive, likewise to the extent of the M_w_ and M_n_ weights. The SEM images revealed that the morphology of the surface films’ appearance changed more drastically after their exposure to the combined thermo- and photo-oxidative treatments. The surface seemed eroded, showing the presence of the inorganic filler of CaCO_3_ as main filler and, at a low content, Mg.

These results suggest that the combined oxidative treatment (thermo- and photo-oxidative) produced degradation of products with a lower molecular weight and greater content of ester and carboxylic groups, which should enhance the LDPE film’s environmental biodegradability.

## Figures and Tables

**Figure 1 polymers-17-00193-f001:**
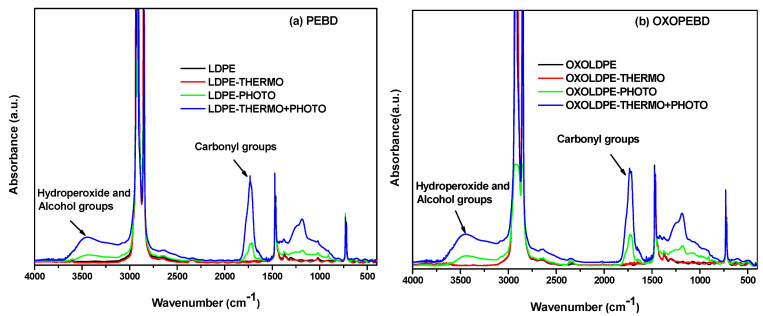
FTIR spectra of LDPE (**a**) and OXOLDPE (**b**) samples after different oxidative tests.

**Figure 2 polymers-17-00193-f002:**
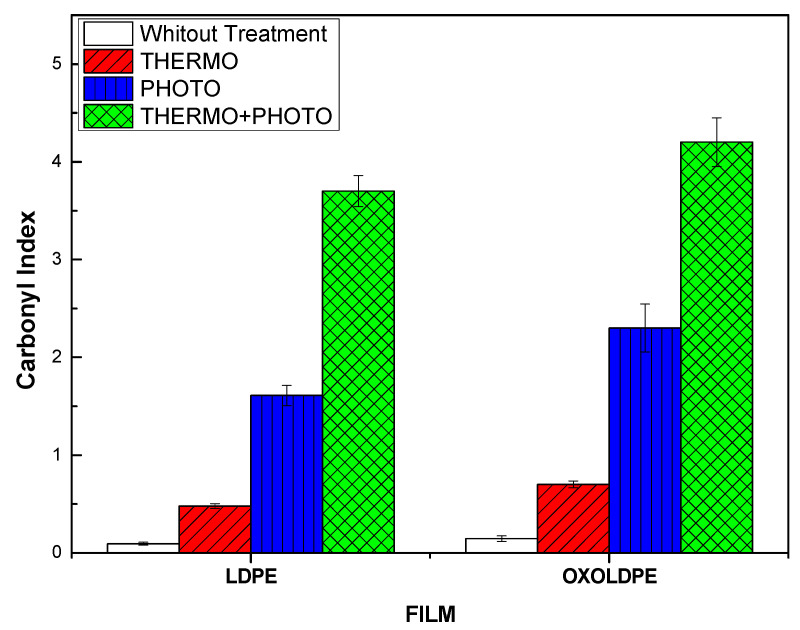
Carbonyl index (CI) of LDPE and OXOLDPE samples after different oxidative tests.

**Figure 3 polymers-17-00193-f003:**
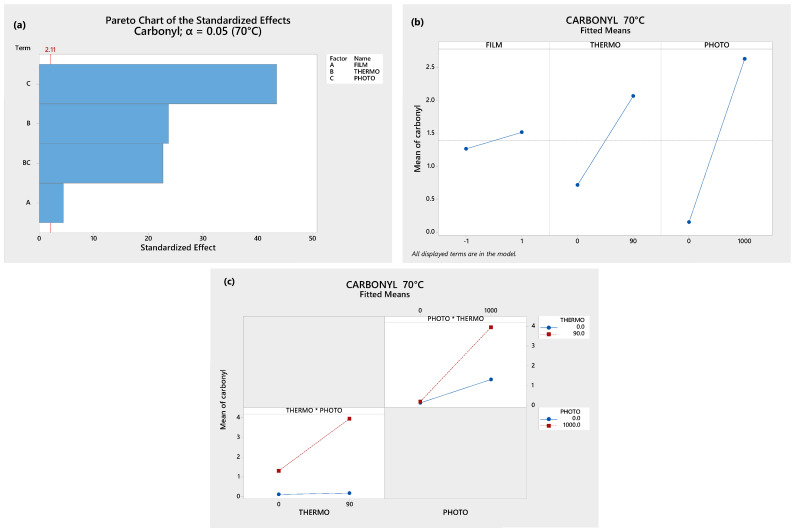
ANOVA results: (**a**) Pareto charts, (**b**) principal effects graph, and (**c**) interaction plot.

**Figure 4 polymers-17-00193-f004:**
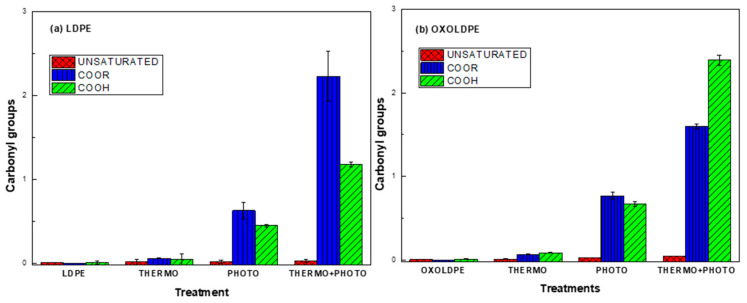
ANOVA analysis of ester and carboxylic group index of LDPE (**a**) and OXOLDPE (**b**) samples after different oxidative treatments.

**Figure 5 polymers-17-00193-f005:**
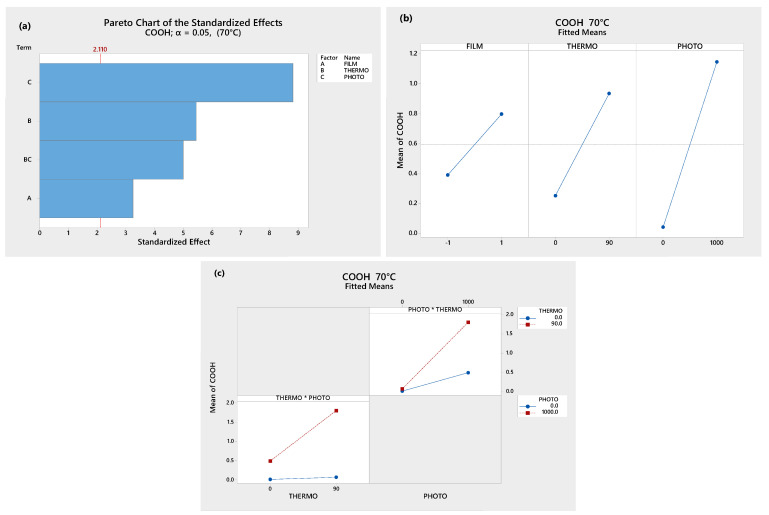
ANOVA analysis of the carboxylic acids (COOH) formed in OXOLDPE samples films after different oxidative treatments: (**a**) Pareto chart, (**b**) principal effects graph, and (**c**) interaction plot.

**Figure 6 polymers-17-00193-f006:**
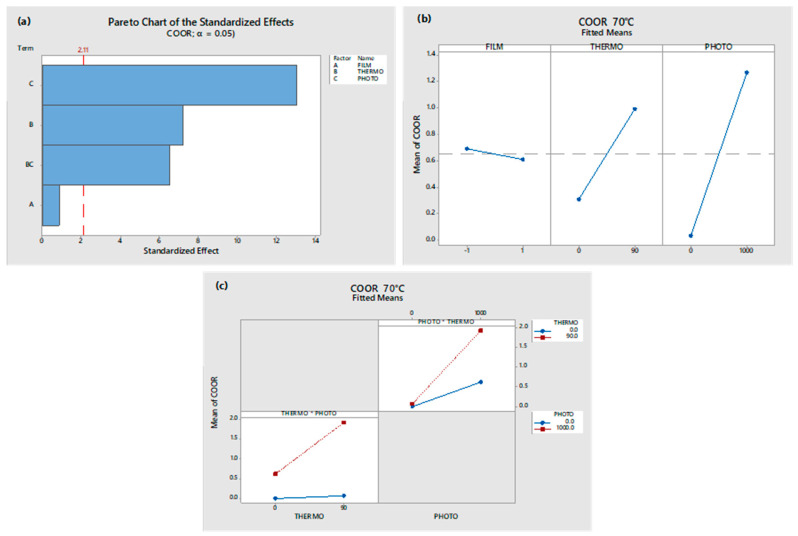
ANOVA analysis of esters (COOR) formed in OXOLDPE sample films after different oxidative treatments: (**a**) Pareto chart, (**b**) principal effects graph, and (**c**) interaction plot.

**Figure 7 polymers-17-00193-f007:**
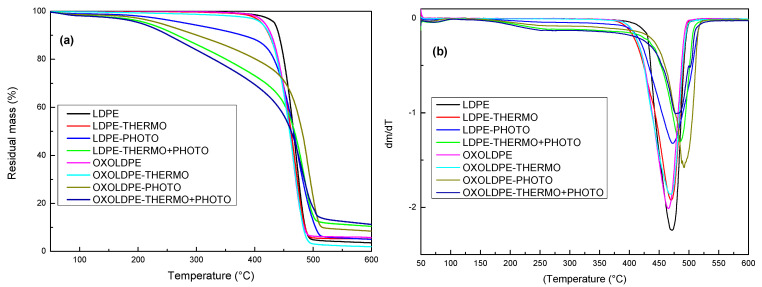
Change in thermal stability of LDPE and OXOLDPE film samples after different treatments: (**a**) residual mass (%) as a function of temperature and (**b**) mass loss rate dm/dT.

**Figure 8 polymers-17-00193-f008:**
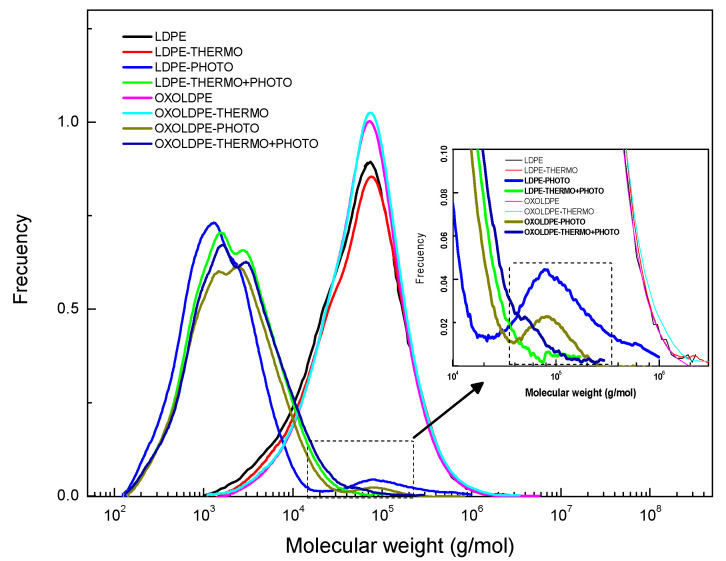
Comparison of the molecular weight distributions obtained by the analysis of the GPC of LDPE and OXOLDPE films subjected to different oxidative treatments.

**Figure 9 polymers-17-00193-f009:**
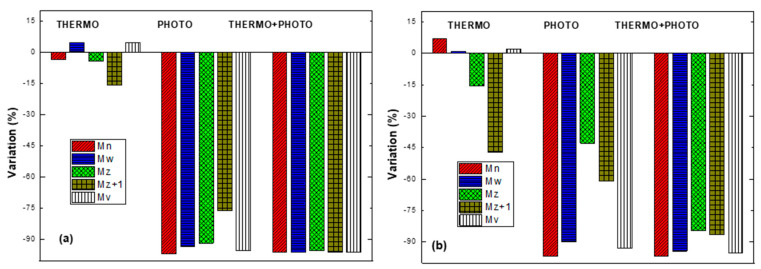
The change in different molecular weights of LDPE (**a**) and OXOLDPE (**b**) films after different abiotic oxidative treatments.

**Figure 10 polymers-17-00193-f010:**
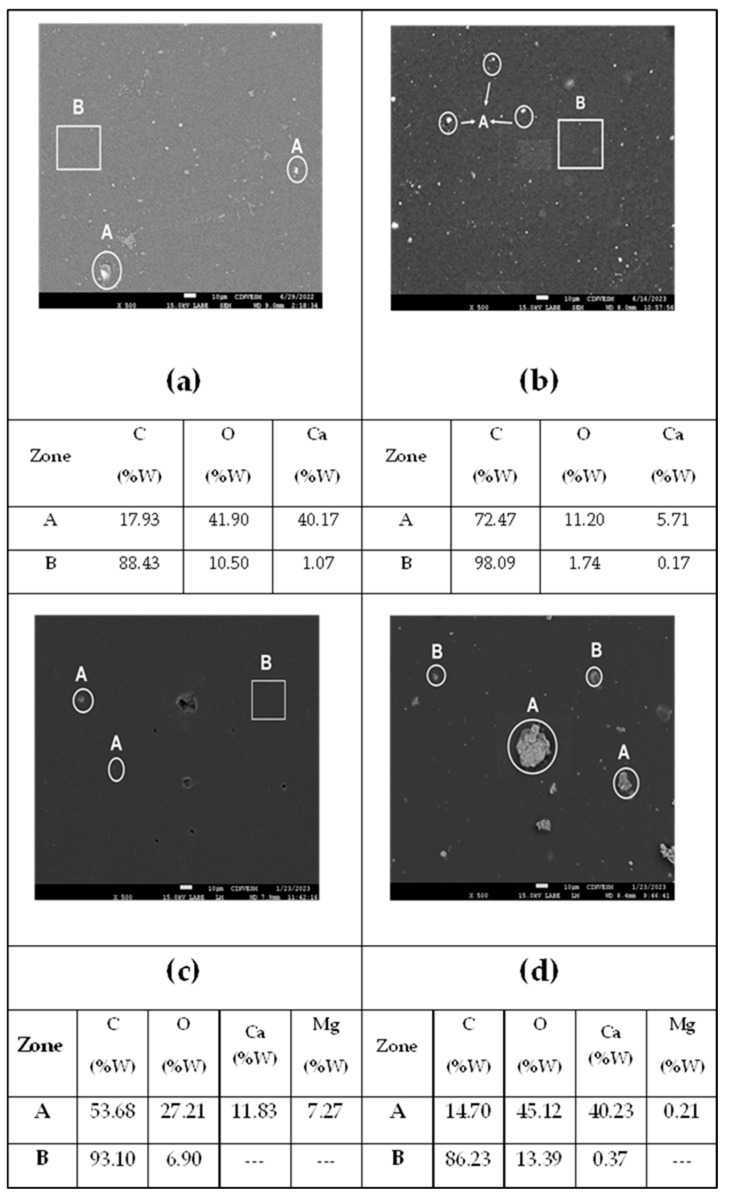
SEM images (x500) of the surface of (**a**) non-treated OXOLDPE film and after different abiotic oxidative treatments (**b**) thermo-oxidative, (**c**) photo-oxidative and (**d**) thermo+photo-oxidative.

**Table 1 polymers-17-00193-t001:** Sample codes.

Sample Code	Description
LDPE	As-received LDPE film (without treatment)
LDPE-THERMO	LDPE film thermo-oxidized
LDPE-PHOTO	LDPE film photo-oxidized
LDPE-THERMO+PHOTO	LDPE thermo-oxidized and then photo-oxidized
OXOLDPE	As-received Oxo-biodegradable LDPE film (without treatment)
OXOLDPE-THERMO	Oxo-biodegradable LDPE film thermo-oxidized
OXOLDPE-PHOTO	Oxo-biodegradable LDPE film photo-oxidized
OXOLDPE-THERMO+PHOTO	Oxobiodegradable LDPE thermo-oxidized and then photo-oxidized

**Table 2 polymers-17-00193-t002:** Crystallinity and crystal size of LDPE and OXOLDPE films based on XRD spectra analysis.

SamplesTreatment	X_XRD_ (%)		L_110_ (nm)	
	LDPE	OXOLDPE	LDPE	OXOLDPE
WITHOUT TREATMENT	54	45	19.5	18.7
THERMO-	65	54	22.1	21.7
PHOTO-	66	73	22.6	19.2
THERMO+PHOTO	75	79	21.3	20.5

**Table 3 polymers-17-00193-t003:** Temperature for selected weight loss of studied LDPE and OXOLDPE film samples after oxidative treatments.

Sample and Treatment	T_10%_ (°C)	T_25%_ (°C)	T_max_ (°C) dm/dT (%)
LDPE	442	453	471/40
LDPE+THERMO	425	444	470/35
LDPE+PHOTO	379	441	472/43
LDPE+THERMO+PHOTO	268	387	485/31
OXOLDPE	429	445	465/40
OXOLDPE+THERMO	426	443	468/35
OXOLDPE+PHOTO	302	435	492/35
OXOLDPE+THERMO+PHOTO	254	366	482/33

**Table 4 polymers-17-00193-t004:** Molecular weight averages of abiotic-treated LDPE and OXOLDPE films.

Molecular Weight Averages	M_n_	M_W_	M_Z_	M_Z+1_	M_V_	M_W_/M_n_
LDPE	24,323	99,048	363,924	1,156,245	83,376	4.1
LDPE-THERMO	23,429	103,476	348,152	971,778	87,195	4.4
LDPE-PHOTO	775	6543	30,668	278,466	3860	8.4
LDPE-THERMO+PHOTO	1000	3945	16,574	46,797	3222	3.9
OXOLDPE	31,256	98,630	298,559	1,046,386	85,564	3.16
OXOLDPE-THERMO	33,462	99,721	252,369	553,030	87,419	3
OXOLDPE-PHOTO	1043	9989	170,790	408,802	6146	9.6
OXOLDPE-THERMO+PHOTO	1042	5559	46,025	139,820	4149	5.3

## Data Availability

The datasets presented in this article are not readily available because the data are part of an ongoing study.

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
