# Peer review of "Impact of Combined Thermo- and Photo-Oxidation on the Physicochemical Properties of Oxo-Biodegradable Low-Density Polyethylene Films"

_polymers, 2025, doi:10.3390/polym17020193_

Round 1
Reviewer 1 Report
Comments and Suggestions for Authors
This paper studied the combined effect of thermo- and photo-oxidation on degradation of LDPE and OXOLDPE. Combined effect is proved to be effective, while OXOLDPE does not show enough pro-oxidant effect.
There are some problems to be answered:
Equation (1) is for all carbonyl species, you can't use it to distinguish the changes of acid and ester. Since C=C peak is not in the carbonyl range, this equation certainly can not be used.
From Fig.1, OXOLDPE series have obviously lower carbonyl peaks than LDPE series, not as stated in the text.
The resolution of Fig.7 shall be improved
In Fig.8, GPC curves of LDPE and OXOLDPE before treatment shall be added.
The title of "4. Discussion" can be deleted, since there are litttle discussions.
In Conclusions, "there was a high concentration of ester groups and carboxylic acids, suggesting that the reaction mechanism is of the Norrish I type, as a consequence of the scission of the polymeric chains being end-of-chain groups", why? There is no data to support this conclusion. If there are mainly end-chain scission, the molecular weight will not drop dramatically. The authors challenge their own results?
Comments on the Quality of English Language
There are many grammar mistakes, the manuscript shall be checked carefully.
Author Response
Reviewer 1
Comment 1: This paper studied the combined effect of thermo- and photo-oxidation on degradation of LDPE and OXOLDPE. Combined effect is proved to be effective, while OXOLDPE does not show enough pro-oxidant effect.
Answer: Thank you for your comment. Indeed, the results of this study show that the combined oxidative treatment is more effective than the presence of the pro-oxidant. The results of the EDS analysis only show the presence of Ca and Mg in the OXOLDPE sample. Calcium has been reported as a moderate pro-oxidant of LDPE by Pablos et al (Pablos, J.; Abrusci, C.; Marín, I. ;López-Marín, J. ;Catalina, F. ;Espí, E.; Corrales, T. Photodegradation of polyethylenes: Comparative effect of Fe and Ca-stearates as pro-oxidant additives. Polym Degrad Stab, 2010. 95(10): p. 2057-2064) and we attribute the observed behavior to that.
There are some problems to be answered:
Comment 2: Equation (1) is for all carbonyl species; you can't use it to distinguish the changes of acid and ester.
Answer:
We thank you again for your comment.
Equation 1 allows us to calculate the global carbonyl index of the samples and not the specific contributions of the different functional groups (ester, carboxylic, keteone, etc) that absorb in that region of the infrared. To clarify how the ester and carboxylic group indices were calculated, we added a paragraph in the experimental section corresponding to the FTIR as follows:
“The carbonyl region was deconvoluted using the Lorentzian non-linear least squares fitting procedure (Origin Pro 9) to calculate the esters (COOR) and carboxylic acids (COOH) contribution. The ester and carboxylic acid indices were calculated using the 1740 cm-1 and 1710 cm-1, normalized with the methylene absorption peak, respectively, like CI.”
Comment 3: Since C=C peak is not in the carbonyl range, this equation certainly cannot be used.
Answer: We added a sentence in the experimental section corresponding to the FTIR as follows: “The unsaturation (C=C) index was calculated using the peak at 990 cm-1 and also normalized with the methylene absorption peak.”
Comment 4: From Fig.1, OXOLDPE series have obviously lower carbonyl peaks than LDPE series, not as stated in the text.
Answer: Thank you for your comment.
In the original figure 1 it appears that the absorption peaks of LDPE are of greater magnitude than those of OXOLDPE. The spectra shown in the original figure 1 are unnormalized and it is known that the absorption of the functional groups depends on the concentration of these and the thickness of the analyzed film. To compensate for the differences in the thickness of the analyzed films, it is customary to normalize the spectra with an internal reference band. In our case we chose the absorption peak at 2020 cm-1 which is reported not to change with degradation (49. Valadez-Gonzalez, A.; Cervantes-Uc, J.; Veleva, L. Mineral filler influence on the photo-oxidation of high density polyethylene: I. Accelerated UV chamber exposure test. Polym Degrad Stab, 1999. 63(2): p. 253-260. https://doi.org/10.1016/S0141-3910(98)00102-5.) to normalize the spectra which are included as the new figure 1.
In our opinion, what is described in the manuscript is more consistent with what is observed in the new figure 1.
Comment 5: The resolution of Fig.7 shall be improved
Answer: New Figure 7 is included with improved resolution.
Comment 6: In Fig.8, GPC curves of LDPE and OXOLDPE before treatment shall be added.
Answer: The new figure 8 is included, which already contains the molecular weight distributions (MWD) of the LDPE and OXOLDPE films without treatment.
Comment 7: The title of "4. Discussion" can be deleted, since there are litttle discussions.
Answer: Thank you for your comment.
Section 4 was removed and section 3 was modified as:
“3. Results and discussions”
Comment 8: In Conclusions, "there was a high concentration of ester groups and carboxylic acids, suggesting that the reaction mechanism is of the Norrish I type, as a consequence of the scission of the polymeric chains being end-of-chain groups", why? There is no data to support this conclusion. If there are mainly end-chain scission, the molecular weight will not drop dramatically. The authors challenge their own results?
Answer: We appreciate your comment.
It is known that Norrish I involves the direct scission of the bond adjacent to an excited carbonyl group, with the formation of two radicals leading to the formation of terminal alcohol and carboxyl acid groups, whereas the Norrish II proceeds through a chain scission without producing radicals leading to the formation of vinylidenes and aldehydes. (Mamin, E.A.; Pantyukhov,P.V.; Olkhov, A.A. Oxo-Additives for Polyolefin Degradation: Kinetics and Mechanism. Macromol 2023, 3,477–506. https://doi.org/10.3390/macromol3030029)
In our opinion, the results shown in this study suggest that the Norrish I mechanism predominates during the photo-oxidation of the films studied since the formation of carboxylic groups predominates over the formation of unsaturated groups.
The indicated paragraph was modified and rewritten as follows:
"Deconvolution in the carbonyl band indicated a high concentration of ester groups and carboxylic acids, suggesting that the Norrish I reaction mechanism predominates during the cleavage of the polymer chains."

Reviewer 2 Report
Comments and Suggestions for Authors
This study investigates the combined effect of thermo-oxidative and photo-oxidative treatments on the physicochemical properties of low-density polyethylene films, with and without oxo-degrading additive. The results show that the combined treatment significantly increases the carbonyl index and crystallinity compared to single treatments. Furthermore, the degradation products of the combined treatments contain more ester and carboxyl groups. Statistical analysis revealed a synergistic effect between thermo- and photooxidative treatments. TGA, GPC and SEM analyzes showed a greater loss of thermal stability, a decrease in molecular weight and notable surface changes, suggesting that the combined treatments improve biodegradability. The work and topic may be of interest to the Polymers Journal's scientific audience; however, some issues need to be resolved before publication.
· The authors should expand the introductory section by adding a part that explains how additives facilitate the fragmentation of materials, which do not degrade completely but break down into tiny fragments that remain in the environment, and what the effects of this degradation are.
· The authors should specify whether the photo-oxidized samples were prepared in triplicate.
· In the GPC methodology the concentration of the sample is not clear.
· As for the graphical part, the authors should improve the quality of the images as they are not easily interpretable. In figures 2, 4, and 9, I would use different colors instead of using various design.
· The authors might consider changing the title to 'Impact of Combined Thermo- and Photo-Oxidation on the Physicochemical Properties of Oxo-Biodegradable Low-Density Polyethylene Films,' as it maintains the focus on the scientific aspects, using a technical formulation that highlights the central theme of the research. Additionally, it is concise and easily understandable, which could attract the attention of researchers or professionals in the field."
All these details should be explained before the manuscript is published.
Author Response
Reviewer 2
This study investigates the combined effect of thermo-oxidative and photo-oxidative treatments on the physicochemical properties of low-density polyethylene films, with and without oxo-degrading additive. The results show that the combined treatment significantly increases the carbonyl index and crystallinity compared to single treatments. Furthermore, the degradation products of the combined treatments contain more ester and carboxyl groups. Statistical analysis revealed a synergistic effect between thermo- and photooxidative treatments. TGA, GPC and SEM analyzes showed a greater loss of thermal stability, a decrease in molecular weight and notable surface changes, suggesting that the combined treatments improve biodegradability. The work and topic may be of interest to the Polymers Journal's scientific audience; however, some issues need to be resolved before publication.
Comment 1: The authors should expand the introductory section by adding a part that explains how additives facilitate the fragmentation of materials, which do not degrade completely but break down into tiny fragments that remain in the environment, and what the effects of this degradation are.
Answer:
Thanks for your comment.
In the Introduction we include a couple of paragraphs and three new bibliographic references concerning the formation of microplastics in the presence of pro-oxidant additives.
"Incorporating oxo-degradant additives can accelerate the degradation process of LDPE since transition metals can act as catalysts for photo- and thermal-degradation processes. The catalytic degradation of polyethylene in the presence of transition metals has been attributed to its ability to generate free radicals on its surface. These radicals later react with oxygen to generate low-molecular-weight oxidation products as oxygen is introduced into the carbon chain as hydroxyls, carbonyls, and peroxides[3-5]
"These materials are called oxo-biodegradable, but there is currently controversy over whether oxo-biodegradable plastics are actually biodegradable and usable as a carbon source by microorganisms. Many studies confirmed that oxo-biodegradable plastics are biodegradable [9-14], while many others consider that they are not [15-20]. This discrepancy has even led the EU to prohibit using such materials [21]. In addition, several studies indicate that oxo-biodegradable plastics tend to generate a more significant amount of microplastics when they degrade in the environment compared to conventional plastics because the formation of these microparticles is closely related to the cleavage of the polymeric chains, the chemical structure of the degradation products formed and the increase in the percentage of crystallinity [22-24]".
- Markowicz, F.; Szymańska-Pulikowska, A. Analysis of the possibility of environmental pollution by composted biodegradable and oxo-biodegradable plastics. Geosci, 2019, 9(11),p 460. https://doi.org/10.3390/geosciences9110460
- Napper I.E.; Thompson, R.C. Environmental Deterioration of Biodegradable, Oxo-biodegradable, Compostable, and Conventional Plastic Carrier Bags in the Sea, Soil, and Open-Air Over a 3-Year Period, Env. Sci. & Tech. 2019 53 (9), 4775-4783. https://doi.org/10.1021/acs.est.8b06984
- Yang, Y.; Li, Z.; Yan, C.; Chadwick, D.; Jones, D.L.; Liu, E., ... & He, W. Kinetics of microplastic generation from different types of mulch films in agricultural soil. Sci.Tot. Environ. 2022, 814, 152572. https://doi.org//10.1016/j.scitotenv.2021.152572
Comment 2: The authors should specify whether the photo-oxidized samples were prepared in triplicate.
Answer: The section 2.2.2 was modified accordingly
2.2.2 Photo-oxidative test
The samples in triplicate were exposed to an accelerated UV aging camera (QUV-SPRAY, UK) and continuously irradiated with UVB-313 lamps (0.71 W/m2) at 60°C for 1000 hours.
Comment 3: In the GPC methodology the concentration of the sample is not clear.
In the section 2.3.4 we added the paragraph:
For this, 20 mg of the sample was weighed and dissolved in 15 ml of 1,2,4-Trichlorobenzene for 1 hour at 170°C. The solution was then filtered through stainless steel filters (0.5 microns).
Comment 4: As for the graphical part, the authors should improve the quality of the images as they are not easily interpretable. In figures 2, 4, and 9, I would use different colors instead of using various design.
Answer: Thank You for your observation.
We modified the Figures 2,4 and 9 in order to improve it.
Comment 5: The authors might consider changing the title to 'Impact of Combined Thermo- and Photo-Oxidation on the Physicochemical Properties of Oxo-Biodegradable Low-Density Polyethylene Films,' as it maintains the focus on the scientific aspects, using a technical formulation that highlights the central theme of the research. Additionally, it is concise and easily understandable, which could attract the attention of researchers or professionals in the field."
Answer: Thank you for your comment.
We find your observation appropriate and believe that the suggested title more clearly reflects the objective and focus of this study.
The title was modified as follows:
Impact of Combined Thermo- and Photo-Oxidation on the Physicochemical Properties of Oxo-Biodegradable Low-Density Polyethylene Films
All these details should be explained before the manuscript is published.

Round 2
Reviewer 1 Report
Comments and Suggestions for Authors
The authors revised Fig.1, why did not explain why the former version seems unreasonable. This makes the revision not serious.
Author Response
Question 1: The authors revised Fig.1, why did not explain why the former version seems unreasonable. This makes the revision not serious.
Answer: Thank you for your comment.
The results and discussion of the FTIR characterization results presented in this work were carried out on normalized infrared spectra; that is, the spectra obtained from the Thermo Scientific Nicolet 8700 spectrometer were divided by the intensity of the absorption band at 2020 cm-1 to compensate for the difference in thickness of the different samples analyzed. This procedure has been recommended to reduce the variability between samples since it is known that the intensity of the absorption bands depends on the thickness of the films.
The band at 2020 cm-1 has been assigned to the asymmetric out-of-plane bending (twisting) of the methylene (CH2) groups in the polyethylene main chain, which is found in the crystalline and amorphous phases of the material. Hence, it has been used as an internal standard.
On the other hand, it is important to note that in the version of the manuscript initially submitted, the spectra shown in Figure 1 were not normalized due to an oversight on our part, for which we sincerely apologize. This caused apparent discrepancies in the analysis of the results.
To improve the clarity of the manuscript, in the materials section we included the sentence “The spectra obtained were normalized using the band at 2020 cm-1 as internal standard before they were analyzed [47-48].”
And in the bibliography section two new references were included:
- Hoekstra, H.D; Spoormaker, J.L; Breent, J; Audouin, L.; Verdu, J. UV-exposure of stabilized and non-stabilized HDPE films: physico-chemical characterization. Polym. Degrad. Stab., 1995. 49: p. 251-262; https://doi.org/10.1016/0141-3910(95)87007-5
48. Graciano-Verdugo A.Z; Peralta E; Gonzalez-Rios H; Soto-Valdez H. Determination of tocopherol in low-density polyethylene (LDPE) films by diffuse reflectance Fourier transform infrared (DRIFT-IR) spectroscopy compared to high performance liquid chromatography (HPLC). Anal. Chim. Act. 2006. 557: p. 367-372. doi:10.1016/j.aca.2005.10.040
